# Antimicrobial Activity of Sunflower (*Helianthus annuus*) Seed for Household Domestic Water Treatment in Buhera District, Zimbabwe

**DOI:** 10.3390/ijerph19095462

**Published:** 2022-04-29

**Authors:** Caston Mahamba, Lobina G. Palamuleni

**Affiliations:** 1Department of Environmental Sciences, College of Agriculture and Environmental Sciences, University of South Africa, Pretoria 0003, South Africa; castonmahamba@yahoo.com; 2Unit of Environmental Science and Management, Department of Geography and Environmental Sciences, Faculty of Natural and Agricultural Sciences, North West University, Mmabatho 2735, South Africa

**Keywords:** antimicrobial, bacteriological, *Helianthus annuus*, seed extract, water treatment, zone of inhibition

## Abstract

Various plants have been used by humans for a very long time, and the uses vary, including food, medicine, toothpaste, dyes, food preservatives, water treatment, and beer brewing, among others. For food preservation and water treatment, the plant must have antimicrobial properties which are biocidal. For this research, extracts were obtained from sunflower (*Helianthus annuus*) seeds. The extracts were assessed for the presence of antimicrobial properties against three groups of bacteria, including faecal coliforms, total coliforms, and *Escherichia coli* (*E. coli*). Dosages of ground sunflower seeds ranging from 0.5 g to 4 g were administered to the three bacterial species and their susceptibilities to the antimicrobial agents were measured and recorded. The results indicate the presence of antimicrobial properties in sunflower. The antimicrobial activities were more effective on *E. coli*, with an average zone of inhibition of 12 mm with a 3 g dosage of sunflower seed extract. This was followed by total coliforms (11 mm) and lastly faecal coliforms (11 mm). These findings suggested that sunflower seeds proved to be potentially effective in treating water against microbial contaminants.

## 1. Introduction

Water is the single most used resource by humans and its availability and quality has assumed enormous global debates and investments [1]. Human activities lead to large amounts of toxic compounds being unleashed into the biosphere at unprecedented rates, sometimes making the environment very inhabitable to most plant and animal species. These toxic wastes, if not well monitored and managed, could possibly exterminate human, plant, and animal life from the face of the earth through their inhalation or consumption [2]. As countries continue to industrialize and the human population continues to grow exponentially, the generation of waste in various forms has also become an everyday challenge for most rural communities, especially in developing countries [3]. In particular, water sources may be adversely contaminated by faecal coliforms through indiscriminate disposal of domestic waste, seepages from pit latrines, as well as inadequately treated wastewater discharged into water systems. The World Health Organization (WHO) guidelines for drinking water quality recommend that faecal indicator bacteria, especially *E. coli* or total faecal coliforms, should not be detectable in any 100 mL drinking water sample [4]. However, it is estimated that 1.8 billion people globally use a source of drinking water that suffers from faecal contamination. Water contamination with coliform bacteria is the main source of waterborne diseases such as gastroenteritis, dysentery, diarrhoea, and viral hepatitis [5].

Before water can be passed into the public supply, it is necessary to remove all potentially pathogenic micro-organisms [6]. Various water treatment methods have been used over the years with varying degrees of success. Most of these methods are quickly losing their relevance, since they cannot be applied on a large scale and also cannot eliminate some of the anthropogenically induced pollutants such as heavy metals and micro-organisms such as faecal coliforms. Since micro-organisms are extremely small, it is not possible to guarantee their complete removal by sedimentation and filtration, therefore the water must be disinfected to ensure its quality. Disinfection is the inactivation of pathogenic organisms, which is the destruction of all micro-organisms present in water [7]. Worldwide, the most common disinfectant used in water supply is chlorine. However, it is not readily available amongst rural poor communities.

Among the various methods of water treatment is phytoremediation, which is a type of bioremediation that uses plants or their tissues and is often proposed for bioaccumulation of metals and other dissolved substances [8]. Phytoremediation has assumed prominence as a low-budget, cost-effective, and environmentally friendly technology for the removal of impurities from water such as alkanes, perchlorates, and metals [9]. The treatment of water, through phytoremediation with respect to perchlorates and chloride compounds, has seen a great deal of success. Even though phytoremediation as a water treatment technique may be successful, its implementation is not very feasible, as there is still much to be studied regarding rates and the probable after effects of microbial activities, as well as treating water on a large scale.

*Helianthus annuus*, known as globally as “the common Sunflower”, is a plant native to North America [10]. In most cases, *Helianthus annuus* is cultivated for its seeds, which contain high amounts of vitamin E edible oil. In addition, *Helianthus annuus* is used in ethano-medicine for treating a number of diseases such as heart disease, bronchial and pulmonary affections, and coughs and colds, and in industries for various products ranging from perfumes to pharmaceuticals [11]. As assumed by Ashbolt [12], most bacterial species are found everywhere, including water bodies, and these are harmful to human health [13]. *Helianthus annuus* has been planted and used traditionally in Zimbabwe as an antimalarial, antiasthma, antioxidant, and antimicrobial agent. The sunflower plant contains several chemicals in its various organs, such as alkaloids and phenolics in the leaves, saponin in the flower parts, polyphenol in the root part containing alkaloids, and fatty acids and tannins in the seeds [14,15,16]. Several studies have shown that the *Helianthus annuus* extracts by organic solvents have potential antimicrobial effects against different Gram (+) and Gram (−) bacterial strains [17]. The antimicrobial activity of methanolic extract of seeds from *Helianthus annuus* was studied against *Bacillus subtilis, Staphylococcus aureus, Salmonella typhi,* and *Vibrio cholera* [18]. The seed extract of *Helianthus annuus* showed high activity against *Salmonella typhi*, moderate activity against *Staphylococcus aureus* and *Vibrio cholera*, and less activity against *Bacillus subtilis* [18]. The studies by [17,18] provide evidence that the properties of sunflower seeds are known to inhibit the growth of bacteria in water. Therefore, the present study aimed to evaluate the antimicrobial properties of sunflower seeds and their effectiveness in inhibiting the growth of selected bacterial sub-divisions, notably faecal coliforms, total coliforms, and *E. coli*.

## 2. Materials and Methods

### 2.1. Description of the Study Area

Figure 1A shows Manicaland, a province in the east of Zimbabwe located along latitude 18.9216° S and longitude 32.1746° E, and has seven districts, namely Buhera, Chimanimani, Chipinge, Makoni, Mutasa, Mutare, and Nyanga.

The province has a total area of 36,459 km^2^ (Figure 1B) and an average population density of 48 people per km^2^. The area receives an annual rainfall amount of about 1098 mm, which is the highest in the country, and average annual temperature is 23 °C [19]. Due to the economic meltdown in the country since the early 2000s, the province has not been spared from the economic turmoil that the country has been experiencing. Due to this economic decline, many municipalities and district councils have not been able to maintain their waste management facilities, water reticulation and treatment, let alone the expansion of the infrastructure to match with the ever-increasing population. The province is the second most populated province in the country, contributing about 13.5% of the national population [20]. The main towns in the province are Mutare, which is the provincial capital, Rusape, Chipinge, and Nyanga. The province is a predominantly a rural area; therefore, the majority of the people do not have piped water but rather rely on borehole water (some of which are unprotected wells) and river water for domestic purposes.

### 2.2. Selection of Water Sampling Sites

Most domestic water sources in most rural parts of Zimbabwe include rivers, dams, wells, and boreholes. For this study, the Mwerahari River and some ten (10) boreholes were identified as sources of domestic water for communities within Buhera District (Figure 1C).

The water samples were taken at three main points along the Mwerahari River, which is the main river and flows across almost the whole length of the district. There are a number of anthropogenic activities along this river. The catchment area of the river is about 172 km^2^ and it receives an average annual rainfall of about 780 mm. The samples were collected systematically, guided by the underlying land uses within the catchment area of the river during three different seasons of the year: summer, autumn, and winter. The sampling sites were at Nerutatanga (close to source), Nyashanu Mission High School, which is along the middle part, and the lower section at Fari, which is close to the confluence with the Save River. Thus, the water samples were labelled as upper, middle, and lower sampling point.

According to the Zimbabwe National Statistical Agency [3], there are 1242 boreholes in Buhera and only about 75% of them are working, which means 931 functional boreholes. In this study, the study area was divided into four political administrative constituencies, which are Buhera South, Buhera North, Buhera Central, and Buhera West. At the time of water sample collection, Buhera South, Buhera North, Buhera Central, and Buhera West had 320, 245, 221, and 145 working boreholes, respectively. From each constituency, a 1% sample of the boreholes was selected. A 1% sample meant that Buhera South had three boreholes, 08, 09, and 10, Buhera Central had two boreholes, 06 and 07, Buhera North had three boreholes, 04 and 05, and Buhera West had two boreholes, 01, 02, and 03. Water samples were collected from these 10 boreholes. For each constituency, random sampling was used to select the boreholes from which water samples were collected. The collected water from the ten boreholes was analysed for biological parameters which included total coliforms, faecal coliforms, and *E. coli*.

### 2.3. Water Sampling Methods

For this study, standard procedures for water sampling were followed, whereby 1 L high-density polyethylene (HDPE) bottles were used to draw water samples from the different identified water sources, because these types of bottles can keep the water samples for a longer time without altering the parameters [21]. The bottles were first rinsed with the same sample water three times before any sample was collected for acclimatization purposes.

Faecal coliform counts were tested using the standard procedure of membrane filtration by standard plate counts [22]. Two samples were taken from each sampling site using sterilised bottles of 200 mL. Samples were transported on ice to the laboratory and stored at 4 °C in a refrigerator and were tested within 24 h. A sample of 100 mL was filtered through a 0.45 μm membrane filter (manufactured by Derbgum, Johannesburg, South Africa) using a Gelman Little Giant pressure/vacuum pump machine (model 13156, Gelman Sciences, Ann Arbor, MI, USA). Membrane filters were aseptically transferred onto m-Endo agar and m-FC agar (Merck, Darmstadt, Germany) to selectively isolate total coliform and faecal coliform bacteria, respectively. The membrane retains the bacteria (total faecal coliform) on its surface and was transferred to an m-Fc agar plate and aerobically incubated for 24 h at 37 °C to allow the bacteria to multiply and form colonies [23,24]. After the incubation, all metallic sheen and typical blue colonies on m-Endo and m-FC agar plates were considered to be total and faecal coliforms, respectively. The colonies were enumerated, and the results were recorded. The m-FC agar plates were stored at 4 °C for further identification of faecal coliforms. Number of colonies was counted and linked directly to faecal bacteriological content of river and borehole water using Equation (1):(1)$$\mathrm{Coliform}\;\mathrm{per}\;100\;\mathrm{mL}=\frac{\mathrm{number}\;\mathrm{of}\;\mathrm{feacal}\;\mathrm{coliforms}\;\mathrm{counted}}{\mathrm{number}\;\mathrm{of}\;\mathrm{mL}\;\mathrm{of}\;\mathrm{sample}\;\mathrm{filtered}}\times 100$$

In order to screen for characteristics of bacteria belonging to the family *Enterobacteriaceae* (for this study *E. coli*), the presumptive faecal coliforms from the m-FC agar plates were isolated and sub-cultured onto m-FC agar using the streaking plate method. Isolates with different colonial morphologies for the summer season were individually revived by sub-culturing on m-FC agar and the plates were incubated aerobically at 37 °C for 24 h to obtain pure colonies [25]. Pure colonies were subjected to PCR identification assays.

For DNA extraction and PCR amplification, the individual pure bacteria colonies were inoculated into 3 mL of Luria–Bertani broth (Merck, Darmstadt, Germany) and incubated at 37 °C for 24 h. Aliquots of 1.5 mL from each overnight culture were transferred into microcentrifuge tubes and vortexed for 2 min to acquire a pellet. Genomic DNA was then extracted using a genomic DNA extraction kit and according to the manufacturer’s instructions (Zymo DNA extraction kit). The eluted DNA was stored at −20 °C and later used for PCR identification tests. Presumptive *E. coli* isolates were screened using *uidA*-specific primers. Primers used in this study are presented in Table 1.

PCR reaction mixtures were prepared in 25 μL volumes comprising of 1 μL of the template DNA, 12.5 μL of PCR master mix, 0.5 μL of both oligonucleotide primers (forward and reverse), and 11 μL of RNase-free distilled water. Thermal cycling for *E. coli* was performed as follows: initial denaturation at 95 °C for 10 min, followed by 35 cycles of 95 °C for 45 s, 59 °C for 30 s, 72 °C for 90 s, and a final elongation at 72 °C for 10 min.

Amplified DNA fragments were resolved by electrophoresis using 2% (*w*/*v*) agarose gel containing 0.1 µg/mL ethidium bromide. The amplicons were visualised under UV light [27]. A Gene Genius Bio-Imaging System was used to capture the images using GeneSnap software (version 6.00.22, Syngene, Synoptics, Cambridge, UK). The images were analysed using GeneTools software (version 3.07.01, Syngene, Synoptics, Cambridge, UK) to determine the presence or absence of the targeted bands, and the images were saved in tagged image file format (TIFF).

#### Statistical Analysis

Since the water samples were collected in triplicate per sampling point, mean and standard deviation were calculated and tabulated. The results obtained from the analysis were compared with Standards Association of Zimbabwe (SAZ) and the WHO drinking water quality standards. Additionally, one-way analysis of variance (ANOVA) was applied to compare the seasonal means of the sampling points and to test for the significant variations between sampling points for each water quality parameter. The statistics were performed at 95% confidence interval and at 0.05% alpha.

### 2.4. Sunflower Seed Extraction

In this study, the common sunflower (*Helianthus annuus*) species, a member of the Asteraceae, the Sunflower family, was used out of the 52 species of sunflowers. Sunflower plants were grown for four months in a garden at Nyashanu High School in Buhera District of the Manicaland Province, Zimbabwe. The laboratory experiment for this study was performed using mature sunflower seeds which were harvested from the garden. Only 2 kg of sunflower seeds were used. The seeds were washed twice, firstly with tap water and then with distilled water, and then dried. The seeds were dried in a Jouan EU28 dry air oven at 65 °C for 24 h. The dried seeds were then homogenized using a standard bench-top blender to a fine powder. Exactly 2 g of the ground seed was then suspended in 40 mL of each of the five solvents—two non-polar solvents (n-hexane and chloroform) and three polar solvents (absolute ethanol, acetone, and cold water)—then allowed to extract the different plant metabolites (phytochemicals) by shaking on an orbital shaker for 24 h at room temperature. Table 2 shows the average zone of inhibition of the sunflower seed extract for the five different solvents namely: chloroform, ethanol, hexane, water, and acetone. Since chloroform had the highest zone of inhibition, it was then selected for the rest of the research.

After the extraction, the liquid extract was filtered out using number 1 Whatman filter paper pore size 11 µm, which presents an advantage in that it has a fairly big dust capacity, has low resistance, and a relatively long service life [28]. The solid residue was discarded. The filtrate was then dispensed into round-bottomed flasks and concentrated by evaporation on a rotary evaporator for 4 h. The resultant solution was used as the sunflower organ (i.e., seed) extract. Before further use, the extracts were sterilized by ultrafiltration using a syringe-based bacteriological filter membrane pore size of 0.45 µm.

### 2.5. Determination of Sunflower Seed Antimicrobial Activities

The Kirby Bauer disc/agar well diffusion assay was used to measure the antimicrobial activities of the sunflower seed extract against selected bacterial isolates: *E. coli*, faecal coliforms, and total coliforms.

#### 2.5.1. Test Isolate Preparation

Exposures were performed on *E. coli*, faecal coliforms, and total coliforms. Before use, these isolates were cultured on nutrient broth and incubated using the Scientific Series 9000 incubator at 37 °C for 12 h. After the incubation, the broth cultures were all standardized to match the McFarland 0.5 bacterial suspension in the tube. For this study, the 0.5 McFarland standard was prepared by mixing 0.05 mL of 1.175% barium chloride dihydrate (BaCl_2_·2H_2_O) with 9.95 mL of 1% sulfuric acid (H_2_SO_4_). This was followed by the preparation of a lawn of each isolate on Mueller Hinton (MH) agar (manufactured by the Sisco Research Laboratories (SRL), Mumbai, India) using a sterile swab. The swab was dipped into the standardized culture, squeezing it on the sides of the test tube to get rid of excess liquid, and then sweeping the surface of the MH agar in all dimensions to obtain an even spread. The plates were left partially open to dry in the vicinity of a gas flame to prevent contamination.

#### 2.5.2. Disc Preparation

Filter paper discs were prepared using a paper puncher. Under aseptic conditions in the biosafety chamber, the sterile discs (diameter 60 mm) were impregnated with 0.2 mL of sunflower seed extract and were put in glass Petri dishes and sterilized by heating in an oven for 24 h at 65 °C. The discs were then covered with the different sunflower seed extracts for each bacterium using the solvents Acetone (A), Chloroform (C), Ethanol (E), Hexane (H), and Water (W), and allowed to soak for at least 30 min, and then they were air-dried (Figure 2).

#### 2.5.3. Application of the Discs

The impregnated discs were then firmly pressed onto the MH agar surface using a pair of sterile forceps. After this, the plates were incubated at 37 °C for 24 h followed by examination for signs of bacterial growth. The presence of a clear zone (zone of inhibition) around a colony was taken as confirmation of a colony being susceptible to the sunflower seed extracts [29]. The effectiveness of the seed extracts was estimated by measuring the diameter of the zone of inhibition. Each antimicrobial assay was performed in triplicate so as to ascertain the authenticity of the procedure and the accuracy of the results [30].

#### 2.5.4. Experimental Implementation

To test the effectiveness of the antimicrobial activity of sunflower seed extracts, the water samples collected from the Mwerahari River and borehole number 06 were subjected to the procedure outlined in Section 2.5.1 and Section 2.5.2 A control experiment was carried out using discs impregnated with only the solvents without the sunflower seed extract. When comparing the antimicrobial activity of the control solvents, the discs with extracts showed greater antimicrobial activity.

## 3. Results and Discussion

### 3.1. Water Quality in Buhera District

#### 3.1.1. Bacteriological Contamination in Borehole Water

Faecal coliforms, 1.2 ± 0.4 cfu/100 mL, 0.7 ± 0.2 cfu/100 mL, and 0.3 ± 0.2 cfu/100 mL for summer, autumn, and spring seasons, respectively, were only detected in water samples from borehole six, which is located in close proximity to a big business centre which has a lot of pit latrines, and might lead to groundwater contamination. There is also a dip tank very close to the borehole, which means a lot of cattle come for dipping every fortnight, leaving a lot of dung around the borehole together with human waste, since there are no toilets for the people managing the cattle. There were no traces of *E. coli* in all the ten samples. According to the WHO guidelines on drinking water [31], all water that contains any coliforms is condemned and users of that facility are advised of the consequences of consuming water from such sources thereof. Given that 90% of the boreholes were faecal coliform free means that villagers of Buhera can use water from their boreholes with a higher level of confidence [32].

#### 3.1.2. Bacteriological Contamination in River Water

Table 3 shows the bacteriological water quality from the three sections of the river. The bacterial load representing the three bacterial species under study reveals higher concentrations of the bacteria in river water than in borehole water. This scenario accrues from the fact that river water is more exposed to human activities, animals, and weather changes that lead to bacterial growth compared to borehole water (groundwater) [33].

Faecal coliforms were detected in all samples from the three sections of river (Table 3). The reasons for the presence of the bacteria in the water were mainly due to the fact that people on both sides of the river do their laundry in the river throughout the year, as well as bathe in the river and water their animals in the river, while human excreta is also washed into the river by runoff from surrounding homes where there are no toilets.

A general trend was shown where the levels of total coliforms detected in the water samples were higher in summer in all sections of the river. This suggests that there is more contamination of water during summer, which is the rainy season [34]. During the rainy season, there is an increase in runoff and more water draining into the river from the catchment’s tributaries, resulting in the accumulation of bacterial load. The average total coliforms found in the water samples was 2.4 ± 0.1 cfu/100 mL, which exceeded the WHO and Standards Association of Zimbabwe (SAZ) standards, an indication that the water was not safe for human consumption. Total coliforms were detected in all seasons and across all sections of the river.

*E. coli* was detected in all samples from all sections of the river (Table 3). The highest numbers were in summer (3.9 ± 0.1 cfu/100 mL) while the lowest were in winter (0.2 ± 0.1 cfu/100 mL), and this could be attributed to the absence of rainfall in winter. Any amount of *E. coli* or coliforms in water renders it unfit for human consumption [35]. The fact that *E. coli* was detected in all samples from all three sections of the river suggests that the water must be treated before consumption. Therefore, the water from the Mwerahari River is not safe for human consumption in any season. According to the Standard Associations of Zimbabwe [36], up to 4% count per 100 mL of total coliforms is accepted, though the water must be used with caution. According to the WHO guidelines on drinking water [31], no traces of *E. coli*, total coliforms, or faecal coliforms can be detected in any 100 mL of water, and once detected the water must be condemned and appropriate treatment must ensue.

The results presented in Table 3 show that significant variation was noted during summer across the three sampling sites for all the three types of bacteria. The variation was due to the input of sewage effluents from the runoff discharges from the catchment area. Domestic animal and human wastes are subsequently washed into the river during rainfall events, and this could likely explain the presence of higher bacterial counts during summer [37]. A similar trend was observed for the samples collected during the autumn season. This could be explained by the onset of the rainy season, with more water draining into the river from the catchment’s tributaries. The seasonal variations in the three types of bacteria were found to be statistically insignificant during the winter season because of low flows in the river and no runoff from the catchment area.

### 3.2. Activities of Sunflower Seed Extracts against Bacteriological Contaminants

#### 3.2.1. *E. coli*

*E. coli* bacteria that were cultured in the laboratory were exposed to sunflower seed extracts in a chloroform solvent at four dosages of 0.5 g, 1 g, 2 g, and 3 g to determine which of the four dosages were more effective in inhibiting the growth of the bacteria. Table 4 shows the various zones of inhibition from the four different dosages.

Previous studies have noted that sunflower seeds contain an array of phytochemicals such as phenols, flavonoids, tannins, alkaloids, saponins, and fatty acids, which are very effective antimicrobial agents [36]. In this study, a dosage of 0.5 g showed the least effect in inhibiting the growth of *E. coli*, with an average zone of inhibition diameter of 7 ± 0.1 mm. The highest zone of inhibition on *E. coli* was recorded with a dosage of 3 g, which had an average zone of inhibition of 13 ± 0.3 mm. From 0.5 g to 3 g the percentage in the zone of inhibition increased by 85.71%, showing a great deal of direct proportionality: thus, the more the dosage, the higher the zone of inhibition [38].

#### 3.2.2. Total Coliforms

Total coliforms were also subjected to sunflower seed extracts at four different dosages in a chloroform solvent. The results shown in Table 4 revealed a pattern where the zone of inhibition was directly proportional to the amount of the dosage; thus, the more the dosage, the higher the zone of inhibition. According to Kay and Fricker [39], total coliforms are by nature a bit less susceptible to antimicrobial agents compared to *E. coli*. The dosages ranged from 0.5 g to 3 g. The average zone of inhibition for 0.5 g was 6 ± 0.5 mm and was the least of the four dosages. The highest zone of inhibition was achieved with 3 g dosage, which stood at 11 mm.

#### 3.2.3. Faecal Coliforms

Faecal coliforms were also subjected to dosages of sunflower seed extracts in a chloroform solvent. Table 3 shows the results that were obtained after measuring the zone of inhibition achieved by each dosage of the seed extract. As already observed with the other two bacterial species, the zone of inhibition was greatest when the dosage was at 3 g, with an average zone of inhibition of 11 ± 0.3 mm.

### 3.3. Effectiveness of Sunflower Seed Extracts against E. coli, Total Coliforms, and Faecal Coliforms

In addition to water treatment being an investment in human health, it is also a huge investment in the wellbeing of the environment and the aquatic species [40]. Amongst the myriad of reasons for water treatment, at the household level, water treatment will add an extra layer of security, since provision of safe and adequate potable water to communities is necessary to prevent acute and chronic health effects [41]. Water treatment aims at removing pesticides, bacteria, viruses, and many more physical, chemical, and biological compounds that find their way into water sources to zero concentrations or at least acceptable levels. Any water that lives to that expectation is considered safe to drink and thus promotes good health.

The three bacteria sub-divisions were treated with different dosages of sunflower seed extract at different time intervals. Figure 3 depicts how different dosages of sunflower seed extracts were able to exhibit different zones of inhibition to the three species of bacteria over different time intervals.

At 0.5 g dosage and at 0 to 10 min, there was no effect and a noticeable zone of inhibition on all the three bacterial species. However, at a dosage of 1 g and after 20 min, a 0.3 mm zone of inhibition was observed only on *E. coli*. A significant zone of inhibition on all the three species was noticed after 1 h with a dosage of 3 g. At this time interval and dosage, the zone of inhibition for faecal coliforms was 4 mm, total coliform was 5. 7 mm, while *E. coli* recorded 6 mm. It was at the hour mark with a dosage of 4 g that total coliforms and faecal coliforms reached a maximum of 11 mm zone of inhibition each, while *E. coli* reached a maximum of 12.33 mm, supporting a submission made by [42] that *E. coli* was more susceptible to the seed extract than the other two microbial pathogens. Maximum zones of inhibition were recorded at 2 h with a dosage of 4 g.

The antimicrobial activity of sunflower seed extracts may be due to the considerable amounts of phenolic compounds present in sunflower seeds [17]. The results obtained from this study are similar to previous studies which revealed antimicrobial activity of sunflower seed extracts against *Staphylococcus epidermis*, *E. coli*, *Pseudomonas aeruginosa*, *Candida albicans*, *Staphylococcus aureus*, and *Proteus vulgaris* [43]. Similarly, a study by [44] in Nigeria using sunflower seeds found that the seed extracts were able to eliminate various bacterial species from water. From the study, it was established that bacterial growth was inhibited with very big inhibition zones: *Salmonella* (15 mm), *E. coli* (21 mm), *P. auregonesa* (25 mm), and faecal coliform (23 mm).

### 3.4. Results for Water Treatment Using Sunflower Seed Extracts

The water quality assessment conducted during the summer season in Buhera was subjected to treatment using sunflower seed extracts with a chloroform solvent. The procedure described in Section 2.4.1 to 2.4.3 was followed. The purpose was to test the effectiveness of the seed extracts in treating borehole water and river water against total coliforms, faecal coliforms, and *E. coli*. The results are shown in Table 5.

The results listed in Table 5 show that the seed extract of 2 g had a 100% success rate in removing all the bacterial load from the borehole water. The success of the seed extract was mainly attributed to the antimicrobial agents contained in seeds, which are fatty acids, saponins, and tannins [45]. Previous studies have reported that these compounds destabilize cell membranes and cuticle collagen in the parasite, but their activity is significantly dependent on the dosage and exposure time [46].

In this study, after the treatment, the microbial levels were below 1 cfu/mL, which is an acceptable level for drinking water standards by the WHO and SAZ [31,32]. Similarly, the seed extracts were able to reduce the bacterial load from samples from the upper section of the river by 100%. In the case of the samples from the middle section of the river, there was 75% success in the first treatment (2 g), which reduced the bacterial load from 4.3 ± 0.4 cfu/100 mL to 0.7 ± 0.2 cfu/mL. After the second treatment with a dosage of 4 g, there was 100% eradication of all the faecal coliform bacteria from the water samples, suggesting that 4 g should be recommended for household water treatment.

Total coliforms detected in the water samples from the river were also subjected to sunflower seed extracts with different doses. There was a 100% effectiveness on samples from the upper section of the river, thus bacterial load was reduced from 2.4 ± 0.1 cfu/100 mL to 0. With the samples from the middle section of the river, there was an 83. 7% reduction in the bacterial load using 2 g of sunflower seed extract. The 100% eradication of bacterial contamination in the water was detected with 4 g of sunflower seed extract. With *E. coli*, the trend was similar to the other microbes, with samples from the upper part of the river showing total eradication of the bacteria with the first treatment. For the samples from the middle and lower section there was a 91.18% and 79.5% reduction in the bacterial load, respectively, while the second treatment reduced the load to 0%. The experiments revealed that depending on the bacterial load in the water, the dosage threshold could range from 2 g to 4 g to ascertain that water is safe for human consumption.

## 4. Conclusions and Recommendations

Despite the massive improvements in water treatment methods that are both faster and more effective, the majority of the rural populations, especially in developing countries, do not have access to conventionally treated water and have to settle with the most rudimentary methods of water treatment or none at all. In this study, the bacteriological parameters of water samples from the ten boreholes and the three sections of the river were tested against the WHO and SAZ water quality guidelines. The results revealed variations across the different sources and seasonal variations were also noted.

The three microbial pathogens, namely *E. coli*, faecal coliforms, and total coliforms, were subjected to extracts from sunflower seeds. The study revealed the effectiveness of environmentally friendly water treatment technologies to meet Sustainable Development Goal number six, which aims to “ensure availability and sustainable management of water and sanitation for all” by the year 2030. The presence of micro-organisms such as faecal coliform bacteria (e.g., *Escherichia coli*) and total coliform bacteria in drinking water is a signal that the water is contaminated with disease-causing pathogens [47]. Therefore, the use of sunflower plants (*Helianthus annuus*) for domestic water treatment as an alternative to conventional and top-notch technologies is considered very necessary for water treatment, especially at the household level among poor rural communities. The research findings show that cheap home-based water treatment technologies can be utilised using locally available resources, which do not need specialised training to administer. However, the study recommends an evaluation of the toxicity of the concentration of the treatment extracts to humans based on in vivo experiments to ascertain their effectiveness and use at the household level.

## Figures and Tables

**Figure 1 ijerph-19-05462-f001:**
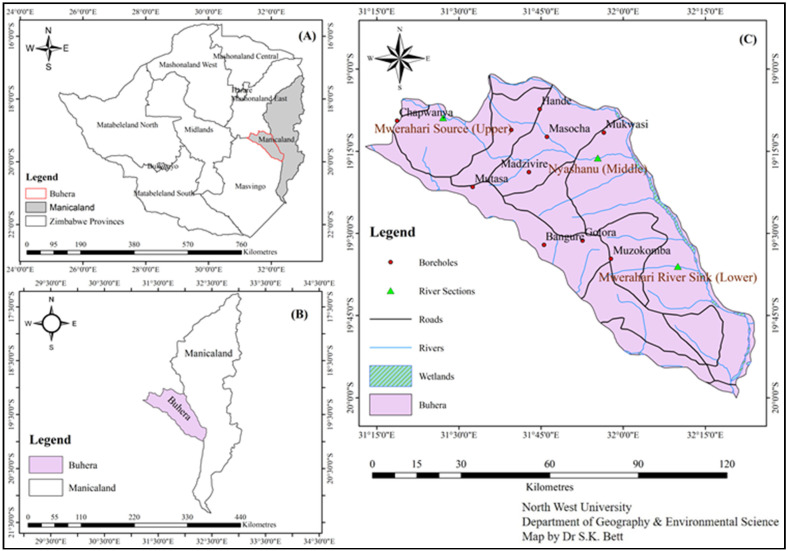
Map of the study area.

**Figure 2 ijerph-19-05462-f002:**
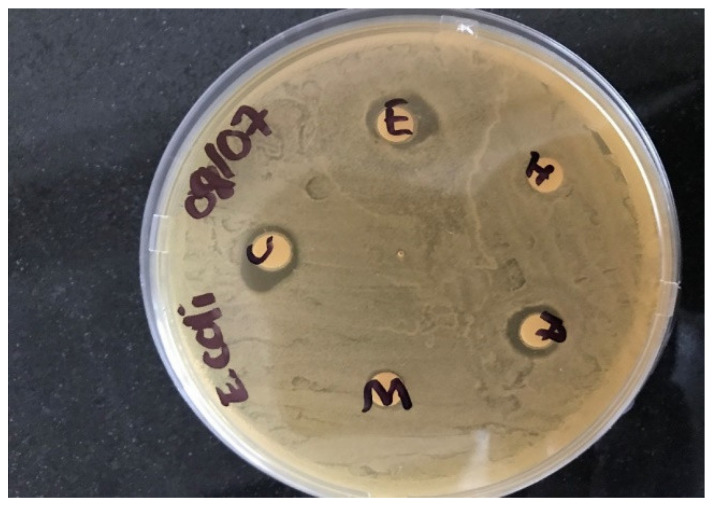
Antimicrobial activities of sunflower seed extracts on different solvents. Source: Researcher (2020).

**Figure 3 ijerph-19-05462-f003:**
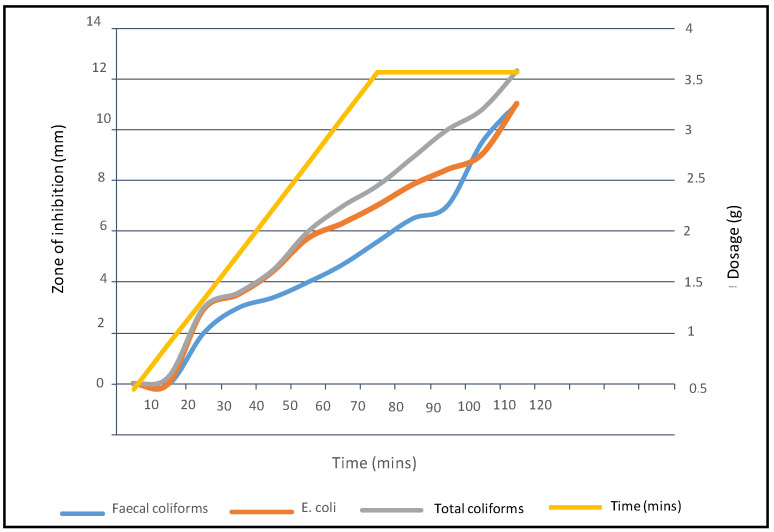
The dosage of sunflower seed extract required to eliminate indicator bacteria per given time intervals and the resultant zone of inhibition.

**Table 1 ijerph-19-05462-t001:** Oligonucleotide primers used for the detection of *E. coli.*

Primer	Primer Sequence (5′ to 3′)	Targeted Gene	Amplicon Size (bp)	Reference
uidA	(F) CTGGTATCAGCGCGAAGTCT (R) AGCGGGTAGATATCACACTC	*uidA*	600	[26]

F = forward; R = reverse.

**Table 2 ijerph-19-05462-t002:** Average zone of inhibition for sunflower seed in five different solvents.

Number of Replication	Solvents for Plant Extract				
	Chloroform	Ethanol	Hexane	Water	Acetone
1	5 ± 0.5 mm	3 ± 0.2 mm	0 mm	4 ± 0.5 mm	3 ± 0.1 mm
2	5 ± 0.1 mm	4 ± 0.0 mm	0 mm	0 ± 0.0 mm	4 ± 0.2 mm
3	5 ± 0.3 mm	3 ± 0.4 mm	0 mm	5 ± 0.2 mm	4 ± 0.5 mm

NB: values are given as mean ± SD (n = 3).

**Table 3 ijerph-19-05462-t003:** Bacteriological parameters in river water samples.

Parameter (cfu/100 mL)	Section of the River	Season						Drinking Water Standards	
		Summer		Winter		Autumn		SAZ	WHO
		Mean ± SD	*p*-Value	Mean ± SD	*p*-Value	Mean ± SD	*p*-Value		
Total coliforms	Upper	2.4 ± 0.1	0.02	0.9 ± 0.2	0.95	1.9 ± 0.3	0.000	0	0
	Middle	4.3 ± 0.4	0.027	1.3 ± 0.2	0.67	2.0 ± 0.1	0.040		
	Lower	1.8 ± 0.2	0.000	2 ± 0.1	0.45	1.1 ± 0.2	0.013		
Faecal coliforms	Upper	2 ± 0.4	0.043	0.6 ± 0.3	0.376	1.3 ± 0.1	0.023	0	0
	Middle	4.1 ± 0.2	0.087	1.2 ± 0.3	1	2.7 ± 0.5	0.000		
	Lower	3.5 ± 0.2	0.09	0.9 ± 0.2	0.71	1.9 ± 0.4	0.068		
*E. coli*	Upper	1.7 ± 0.2	0.01	0.2 ± 0.1	0.665	0.6 ± 0.3	0.025	<1	0
	Middle	3.4 ± 0.1	0.000	1.1 ± 0.4	0.999	1.7 ± 0.4	0.074		
	Lower	3.9 ± 0.1	0.004	0.9 ± 0.3	0.833	1.1 ± 0.2	0.000		

**Table 4 ijerph-19-05462-t004:** Activities of sunflower seed extracts against bacterial species.

Bacterial Species	Dosage (g)			
	0.5 g	1 g	2 g	3 g
*E. coli*	7 ± 0.2 mm	9 ± 0.2 mm	11 ± 0.3 mm	13 ± 0.3 mm
Total coliforms	6 ± 0.5 mm	7 ± 0.7 mm	9 ± 0.6 mm	11 ± 0.4 mm
Faecal coliforms	7 ± 0.1 mm	8 ± 0.2 mm	10 ± 0.3 mm	11 ± 0.3 mm

NB: values are given as mean ± SD (n = 3).

**Table 5 ijerph-19-05462-t005:** Effectiveness of sunflower seed extract on the three selected bacteria during summer season.

Borehole Number	Parameters (cfu/100 mL)					
	First Treatment (2 g)			Second Treatment (4 g)		
	Total Coliform	Faecal Coliform	*E. coli*	Total Coliform	Faecal Coliform	*E. coli*
06	0 (1.2 ± 0.4)	0 (0)	0 (0.5)	0	0	0
Section of River						
Upper	0 (2.4 ± 0.1)	0 (2 ± 0.4)	0 (1.7 ± 0.2)	0	0	0
Middle	0.7 ± 0.2 (4.3 ± 0.4)	1.0 ± 0.1 (4.1 ± 0.2)	0.3 ± 0.2 (3.4 ± 0.1)	0	0	0
Lower	0 (1.8 ± 0.2)	0.4 ± 0.3 (3.5 ± 0.2)	0.8 ± 0.1 (3.9 ± 0.1)	0	0	0

NB: values in parenthesis represent raw water microbial concentrations during summer season; values are given as mean ± SD (n = 3).

## Data Availability

The data presented in this study are available in this article.

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
