# Peer review of "Antimicrobial Activity of Sunflower (Helianthus annuus) Seed for Household Domestic Water Treatment in Buhera District, Zimbabwe"

_ijerph, 2022, doi:10.3390/ijerph19095462_

Round 1

Reviewer 1 Report

This study aimed to investigate the evaluation of the use of sunflower seed to remove the three microbial pathogens namely; E. coli, faecal coliform, and total coliforms from water.

The results obtained from this study can be used by the researchers working on this subject, and it can be a key study in the literature.

Besides these, the manuscript contains good experimentation and reliable results and is written well.

On the other hand, there are some mistakes in the writing of the paper.

And, the paper needs minor correction/revision.

Please see the attached file for my corrections/comments. on the manuscript.

Additionally, please consider the following points:

  • Based on my corrections seen in the attached file, make the corrections.
  • Page 2, Line 84: These studies… which studies? be specific in terms of citations.
  • Page 3, Line 96: The small and big pictures in Figure 1 are not clear. Provide better quality pictures. Not easy to read the map.
  • Page 3: Line 122: Don’t repeat the sentences, and delete duplicates.
  • Page 4: Line 159: Add the equation number.
  • Page 8: Check Table 3.
  • Page 10: Increase the quality of Figure 3.
  • Page 11: Check Table 5.

Please consider all corrections/comments given in the review report to improve the quality of the paper.

Overall, the paper can only be accepted after minor revision to be considered to be published in the journal of IJERPH.

Author Response

Reviewers comments attached

Reviewer 2 Report

Dear Authors,

you have improved your paper and now it is more clearer. 

Some editing error have been found and I suggest you to read your paper once again carefully.

Author Response

Comments to reviewer 2 attached

Reviewer 3 Report

All points have been corrected.

Author Response

Reviewer 3 had no comments for this round

This manuscript is a resubmission of an earlier submission. The following is a list of the peer review reports and author responses from that submission.

Round 1

Reviewer 1 Report

In this study, the authors investigated the use of sunflower seed to remove the three microbial pathogens namely; E. coli, faecal coliform, and total coliforms from water.

This study is interesting and can be used as a reference for readers working on this subject.

The manuscript is well written and presented, the authors presented good experimentation and reliable results.

However, as a reader, I would like to see the detailed literature or discussion in the manuscript.

Otherwise, this paper will look like a technical note rather than a research article.

By the way, there are some minor corrections.

Please see the attached file where you can find my corrections/comments on the manuscript.

Please consider the corrections/comments to improve the quality of the paper.

Then, the manuscript can be accepted to be published in the journal of IJERPH.

Reviewer 2 Report

This manuscript is dealing with sunflower seed extract use for microorganisms’ inhibition. This could be a really interesting paper to read but at the end a lot of questions are arising and I feel confused.

Line 17 It is 0.5 g

Lines 20-21 Space between number and unit is missing (correct it throughout the manuscript and also choose weather it is small or capital letter in units (ml vs. mL)

Line 29 Full stop is missing

Line 69 As assumed by (Add author) [11], …

Introduction: Please provide some references dealing with microorganisms inhibition with sunflower plant.

Since you have collected the samples during summer, autumn and winter, why didn’t you collected them during spring too?

Line 114 According to Author [3], …

Line 118 Buhera West had…

Subsection 2.3. Water sampling – I suggest to rename this subsection into methods or similar.

Lines 137 “…was filtered through a 0.45 μm” finish the sentence and provide the producer of the membrane.

Lines 142-143 I suggest to write the equation with Word “Equation” tool

Please write more info about the agar (producer etc.) used and also the explanation of m-FC agar.

What kind of PCR equipment was used? Please specify.

Subsection 2.4. It would be great if you can specify how much extract you have obtained from the 2 g odžf seeds.

In the materials and methods part more information about equipment and chemicals must be provided!

Table 1 is unclear.

Line 198 Usually dry sterilization is performed at 160 °C for 2 h, and at 180 °C for 1 h. I doubt that you could sterilize anything at 100 °C.

Line 200 Air dried? Could you please specify whether it was dried in microorganism’s free environment? How much of the sunflower seed extracts was used for the E. coli tests?

Figure 2 must be renamed.

Line 216 spaces between units are missing and “0.3”

Subsection 3.1.1. The results are missing (figure/table?)

Did you have a control during the experiments?

Table 2 Decimal dots instead decimal comma. The same in Tables 3 and 4.

In the discussion part (subsection 3.1.2.) the results must be discussed. The average “number CFU/100 mL) is not sufficient.

Lines 250-253 are repeated (see lines 234-238)

Line 254 “Highest figures”?

Line 256 “The fact that E. coli was detected in all samples from all three sections of the river suggests that the water must be treated before it can be consumed” Isn’t it expected since it is a borehole? The physical and chemical parameters, as well as the microbiology in such waters (samples) change from hour to hour and is to be expected that it is not permissible for human consumption (especially since the infrastructure is not established or it does not work properly)

Lines 261-263 are also repeated.

Please write E. coli in italic, as well as other microorganisms (Line 322-323)

I don’t really understand how you have performed the experiments of microorganism’s inhibition with the sunflower seed extract? (Subsection 3.4.). Maybe I missed some part?

Have you considered how would you use this extract for domestic purpose? And what after the water treatment (a filtration is needed or at least sedimentation of the extract)?

The presentation of results and discussion must be improved and maybe the entire manuscript reorganized to make it more clear.

The title of the manuscript is not really supported by the results. 

Reviewer 3 Report

In this manuscript, the authors researched the anti-microbial properties of sunflower seed for inhibition of microorganisms in household domestic water. Finding of authors are thought to be of some use to the water treatment fields. However, I think it will be a better paper if following are revised.

Revision points;

  1. Materials and Methods

- line 164: Provide the variety and collection timing of sunflower seeds.

- line 167: Provide the sunflower seed drying method (method, equipment, time, temperature, etc.) and moisture content of final products.

- Table 1: Express the mean and statistical significance. Please indicate the units of the numbers. How did the control be treated?

- line 189: Write the exact time rather than the expression ‘overnight’.

- line 191: Mueller Hinton agar – Manufacturer, City, Country

- line 197: Mark the diameter of the disc

  1. Results and discussion

- Table 2, 3: Repeated number of experiments, statistical significance. Why did 8.5mm and 11.33mm represent decimal places and other data as integers? need for unification

- Figure 3: What does the yellow line represent? It’s hard to understanding as a whole.

Although the number of test strains decreases when treating sunflower seed extracts, it is necessary to review the sensual changes in water when added.

In addition, overall, statistics on the results are needed.

Author Response

Comments to reviewer 1, reviewer2 and reviewer 3 attached
